# Reconstruction of Lamb Wave Dispersion Curves in Different Objects Using Signals Measured at Two Different Distances

**DOI:** 10.3390/ma14226990

**Published:** 2021-11-18

**Authors:** Lina Draudvilienė, Olgirdas Tumšys, Renaldas Raišutis

**Affiliations:** 1Ultrasound Research Institute, Kaunas University of Technology, K. Baršauskas St. 59, LT-51423 Kaunas, Lithuania; olgirdas.tumsys@ktu.lt (O.T.); renaldas.raisutis@ktu.lt (R.R.); 2Department of Electrical Power Systems, Faculty of Electrical and Electronics Engineering, Kaunas University of Technology, Studentu St. 48, LT-51367 Kaunas, Lithuania

**Keywords:** ultrasonic Lamb wave, non-destructive testing, non-homogeneous material, signal processing, Lamb wave phase velocity, dispersion curve reconstruction

## Abstract

The possibilities of an effective method of two adjacent signals are investigated for the evaluation of Lamb waves phase velocity dispersion in objects of different types, namely polyvinyl chloride (PVC) film and wind turbine blade (WTB). A new algorithm based on peaks of spectrum magnitude is presented and used for the comparison of the results. To use the presented method, the wavelength-dependent parameter is proposed to determine the optimal distance range, which is necessary in selecting two signals for analysis. It is determined that, in the range of 0.17–0.5 wavelength where $\delta_{c_{ph}}$ is not higher than 5%, it is appropriate to use in the case of an A_0_ mode in PVC film sample. The smallest error of 1.2%, in the distance greater than 1.5 wavelengths, is obtained in the case of the S_0_ mode. Using the method of two signals analysis for PVC sample, the phase velocity dispersion curve of the A_0_ mode is reconstructed using selected distances *x*_1_ = 70 mm and *x*_2_ = 70.5 mm between two spatial positions of a receiving transducer with a mean relative error $\delta_{c_{ph}}=2.8\%$, and for S_0_ mode, *x*_1_ = 61 mm and *x*_2_ = 79.7 mm with $\delta_{c_{ph}}=0.99\%$. In the case of the WTB sample, the range of 0.1–0.39 wavelength, where $\delta_{c_{ph}}$ is not higher than 3%, is determined as the optimal distance range between two adjacent signals. The phase velocity dispersion curve of the A_0_ mode is reconstructed in two frequency ranges: first, using selected distances *x*_1_ = 225 mm and *x*_2_ = 231 mm with mean relative error $\delta_{c_{ph}}=0.3\%$; and second, *x*_1_ = 225 mm and *x*_2_ = 237 mm with $\delta_{c_{ph}}=1.3\%$.

## 1. Introduction

Ultrasonic Lamb waves and their possibilities of use in various industrial fields are discussed constantly, and research related to their application has been conducted in an effective manner for the past many years. The use of such waves in structural health monitoring (SHM) and non-destructive testing (NDT) is one of the promising tools for the inspection of composite structures. Based on the carried out investigations, several main useful features of Lamb waves can be named, such as the high sensitivity to the properties of the material, sensitive for cracks at different depths, and the ability of propagation over the entire thickness of the object in quite a long distance (up to 100 m) with relatively low attenuation, etc. [1,2,3,4]. Consequently, due to the named useful features of such waves, the spectrum of their applications grows constantly and has become increasingly significant in various industrial fields. However, the unusual properties of these waves, such as dispersion phenomenon (the velocity as a function of frequency), infinite number of modes, the convergence of modes, modes interlaced, mode splitting due to edges, and other factors [2,5,6,7], create complications in the analysis of receiving signals and, at the same time, limit the applicability of such waves. The dispersion phenomenon, that leads to phase and group velocities, is frequency-dependent, and affects the variation of signal amplitude is named as one of the main limitations of such waves [6,8,9,10]. However, the other problem that complicates the application of the Lamb waves is the infinite number of modes and their convergence. At low frequencies, there are only two dispersion modes of propagating Lamb waves: asymmetric A_0_, and a symmetric S_0_ mode that propagate at different velocities, and the identification of them is a simple task. Therefore, they are most commonly used in various applications [11]. With the frequency increase, the number of modes increases and separation and identification of these regimes is still a difficult task [12].

Multiple methods have been used for signal analysis: two-dimensional Fast Fourier Transform (2D-FFT), Short-Time Fourier Transform (STFT), Wavelets Transform (WT) and its variations, matching pursuit algorithms, Wigner-Ville Distribution (WVD) and its variations, etc. [1,13,14,15,16,17,18], or signal processing methods used for phase velocity and wavenumber estimation [19,20]. However, the named methods require signals collection at many points along the wave propagation path for the reconstruction of dispersion curves of Lamb waves. Therefore, new methods are being developed and proposed that are simple and easy to use, but at the same time provide the necessary information quickly.

The method, which enables reconstructing the phase velocity dispersion curves of Lamb waves using only two adjacent signals, was presented by Draudviliene et al. [21]. The mathematical verification and the experimental research using signals propagating in the homogeneous aluminium plate were performed and presented in that work. The comparison of the obtained results with the Semi Analytical Finite Element (SAFE) method provided very good opportunities and possibilities to use this method for the quick and accurate evaluation of the phase velocity dispersion of the Lamb waves. Therefore, the next step of the research is to perform the proposed method verification and suitability to use it in different non-homogeneous materials and objects of variable complexity. Two different objects, the thin polyvinyl chloride (PVC) film and the large multi-layered wind turbine blade (WTB), are selected for the investigation. The selected objects should show the wide range of applications of the Lamb waves, and the possibility to use them for various industrial applications and as a prevention tool. The use of Lamb waves for waste prevention is one way to address the global challenges of plastic and CO_2_ pollution, and thus contribute to sustainability and a healthy lifestyle.

Thus, the first selected sample used in the study is PVC. The PVC material is widely used for building and construction product manufacture, packaging, agriculture, and other fields. Based on European Union (EU) data provided in [22], it is estimated that approximately 5 million tonnes of PVC was produced, and this accounts for about 10 percent of total plastic production in Europe alone in 2019. Therefore, recently, not only the sustainable management of plastic waste but also the sustainable production of plastic products [23,24], has become a frequent topic of discussion. Thus, the failures occurring during the processing are named as some of the main causes of some defects in the plastic extrusion process [25,26], whereas the Lamb waves are very sensitive to the changes in material, therefore, by using such waves, the thickness change could be determined and likewise, product quality and the required amount of the plastic pellet or powder quantity could be ensured. Therefore, the ultrasonic testing methods based on Lamb waves are named as one of the ways which could be used in the plastic extrusion process for quality control [27] and, at the same time, to contribute to sustainability.

The second selected sample is the WTB. Wind energy is said to be one of the most basic energy sustainability alternatives, therefore, the world is moving rapidly towards renewable energy, and the number of wind farms is constantly growing [28,29,30]. However, with the increasing number of wind farms, the world is facing an ever-increasing number of turbine blades being ejected [31]. The main reasons for this are various defects that occur in the production of WTBs, and the service life can affect the development of damage under fatigue loads [32,33]. Therefore, it is highly important to use techniques that would identify defects and have the ability to assess their location and size during such WTB production [34].

Therefore, the reliable and fast methods for testing, verification, and monitoring of different kind of objects are necessary to be proposed. Application of ultrasonic Lamb waves and evaluation of phase velocity dispersion is one of the ways to solve the named tasks and could be used for the operational security and failure prevention of WTB.

The aim of the presented work is to investigate the possibilities of the method of two adjacent signals analysis to reconstruct the phase velocity dispersion curves of Lamb waves propagating in two different types of objects: PVC film, and a WTB sample possessing a complex construction. To achieve this goal, it is necessary to develop a new algorithm based on peak analysis of the spectrum magnitude of two-dimensional Fast Fourier transform (2D-FFT) data, propose the wavelength-dependent parameter for determination of the optimal distance range which is necessary for selecting two signals for analysis, and to perform comparison of the results obtained using both methods by calculating the mean relative error.

The paper is organized as follows: Section 2 describes two different methods for reconstruction of the dispersion curve segment (method of two signals analysis and 2D-FFT method). The procedure for the performance investigation of methods is presented, and a comparison of both methods is performed for numerically simulated signals. The experimental investigation of different objects (polyvinyl chloride film and WTB) is presented in Section 3. Finally, the conclusive statements are presented in Section 4.

## 2. Methods and Numerical Investigation of Performance

### 2.1. Method of Two Signals Analysis

The comprehensive algorithm of the signal processing method has been presented in [21]. Therefore, only a brief description of the algorithm with the main steps and equations is presented below.

Two adjacent signals $u_{x1}\left(t\right)$ and $u_{x2}\left(t\right)$ at different distances $x_{1}\left(t\right)$ and $x_{2}\left(t\right)$ are measured. The time values (*t*) correspond to the measured distances *x*_1_ and *x*_2_.

The frequency spectra of both measured signals are calculated according to:(1)$$U_{x1}\left(f\right)=\mathrm{FT}\left[u_{x1}\left(t\right)\right],\;U_{x2}\left(f\right)=\mathrm{FT}\left[u_{x2}\left(t\right)\right],$$
where FT is the Fourier transform.

The frequency spectra are filtered using the formed filter packet:(2)$$U_{x1,k}\left(f\right)=U_{x1}\left(f\right)\cdot B_{k}\left(f\right),\;U_{x2,k}\left(f\right)=U_{x2}\left(f\right)\cdot B_{k}\left(f\right).$$
where $B_{k}\left(f\right)=e^{4\ln\left(0.5\right){\left(\frac{f-f_{L}-\left(k-1\right)df}{\Delta B}\right)}^{2}}$ represents the frequency response of *k*-th bandpass filter, *k* = 1,2,...*K*, *K* is the total number of filters, *f*_L_ and *f*_H_ defines the frequency ranges in which the central frequencies of the filters are varied; Δ*B* is the filter bandwidth; and $df=\frac{f_{H}-f_{L}}{K-1}$ is the step in the frequency domain between central frequencies of two neighbouring filters.

The frequency spectra are reconstructed into the time domain using the inverse Fourier transform IFT:(3)$$u_{x1,k}\left(t\right)=\mathrm{IFT}\left(U_{x1,k}\left(f\right)\right),\;u_{x2,k}\left(t\right)=\mathrm{IFT}\left(U_{x2,k}\left(f\right)\right),$$

The envelopes for reconstructed signals are determined:(4)$$e_{x1,k}\left(t\right)=\mathrm{HT}\left[u_{x1,k}\left(t\right)\right],\;e_{x2,k}\left(t\right)=\mathrm{HT}\left[u_{x2,k}\left(t\right)\right],$$
where HT is the Hilbert transform. The peak values of the envelopes are determined, due to the higher concentration of the signal energy and the better signal-to-noise ratio. According to this, the appropriate time delays and closest zero-crossing time instants of the signal waveform segments are determined.

The phase velocity is calculated according to:(5)$$c_{ph,ki}=\frac{x_{2}-x_{1}}{t_{x2,ki}-t_{x1,ki}},$$
where $t_{x1,ki}$ and $t_{x2,ki}$ are the zero-crossing time instants of the reconstructed signals ($u_{x1,k}\left(t\right)$, $u_{x2,k}\left(t\right)$), *i* = 1 ‒ *N* is the number of the zero-crossing instants in each signal, and *N* is the total number of zero-crossing instants.

The equivalent frequencies are calculated:(6)$$f_{ki}=0.125\cdot \left(\frac{1}{t_{x1,k\left(i+1\right)}-t_{x1,ki}}+\frac{1}{t_{x1,ki}-t_{x1,k\left(i-1\right)}}+\frac{1}{t_{x2,k\left(i+1\right)}-t_{x2,ki}}+\frac{1}{t_{x2,ki}-t_{x2,k\left(i-1\right)}}\right)$$

The phase velocities dispersion curve segment is the set of the phase velocity values Equation (5) related with the frequency values Equation (6) $\left(c_{ph,ki},\;f_{ki}\right)$.

Based on the bandwidth of the received signal and the number of the selected filters, the set of points of the extracted dispersion curve is determined. The received signal depends on the bandwidth of the excited transmitting and receiving transducers, the frequency and waveform of the generated signal, and the characteristics of the object are investigated [19]. Based on these parameters, the points of the dispersion curves for each A_0_ and S_0_ modes are obtained.

The phase velocity dispersion estimation is performed using a method of analysis of two adjacent signals at different distances. The method mathematical verification was conducted using simulated signals of the fundamental modes of Lamb waves in a homogeneous aluminium plate. The segments of the phase velocity dispersion curves were obtained in two different frequency ranges of 300 kHz and 1 MHz and compared with the SAFE method. The mean relative error did not exceed 1% for both asymmetric and symmetric modes [21].

The experimental research was performed using signals of Lamb waves propagating in a homogeneous aluminium plate and a non-homogeneous glass fibber reinforce plastic (GFRP) plate. A comparison of the results with the SAFE method showed that in the case of an aluminium plate, the average relative error of the two modes does not exceed 1%. A comparison of the obtained results in the case of GFRP plate with the 2D-FFT method showed a good coincidence [19]. In addition, the relative systematic errors and the expanded uncertainty of the phase velocity A_0_ mode were calculated using different filters, which confirmed the proposed reliability of the method for the GFRP plate [35].

In order to perform the verification of the presented method, a comparison with other methods should be performed. However, due to the fact that important information such as the object elastic constants, geometry and/or velocities of the transverse and longitudinal waves are unknown, the analytical methods SAFE [36] or DISPERSE [37] cannot be used. For that reason, the validated 2D-FFT method is selected and used for comparison with the obtained results.

### 2.2. 2D-FFT Method

The wave propagating along with the object under investigation and characterized by the distance and time can be transformed into the wavenumber and frequency space using the 2D-FFT technique:(7)$$H\left(k,\omega \right)=\iint_{-\infty }^{\infty }u\left(x,t\right)\cdot e^{-j\left(k\cdot x+\omega \cdot t\right)}dxdt,$$
where *x*—coordinate, *t*—time, *ω*—angular frequency, and *k*—wavenumber.

The phase velocity is the ratio of angular frequency *ω* and wavenumber *k.* It can be presented in the 2D space of wavenumber and frequency [38]:(8)$$c\left(\omega \right)=\frac{\omega }{k},\;\omega =2\cdot \pi \cdot f.$$

The outcome of 2D-FFT technique is an image or 2D data array. However, in order to compare different signal processing methods being used for segment estimation of phase velocity dispersion curve, it is necessary to operate with appropriate velocity values of the appropriate Lamb waves mode (e.g., A_0_) obtained at particular frequencies. In practice, after the processing of 2D FFT data, the result of obtained phase velocities is 2D data array *C*. At each frequency value *f*_m_, there is a set of frequency-dependent phase velocity values *c*_i_(*f*_m_) possessing particular energy values of spectrum magnitude, *i* = 1...*N*, *m* = *m*_1_...*m*_2_, where *m*_1_ and *m*_2_ are sample values of frequency within the bandwidth of ultrasonic transducers in a range from *f*_1_ up to *f*_2_.

Therefore, it was proposed to estimate values of phase velocities of the A_0_ mode from 2D-FFT image at maximum energy, applying the peak detection of 2D spectra magnitude at particular frequencies. Such detection of *c*_i_(*f*_m_) peaks at each frequency value is performed within the selected frequency bandwidth from *f*_1_ up to *f*_2_:(9)$$c_{m}\left(f_{m}\right)=\max\left(C(f_{m},c_{l}\left(f_{m}\right))\right),$$
where *l* = 1 … *L,* and *L* = number of samples of phase velocity values.

Further, in order to reduce the influence of the natively blurred shape of 2D spectra magnitude to the results of peaks being detected and avoiding the related undesirable effect of slightly scattered values of the obtained phase velocities (*c_m_*(*f_m_*)), the second order polynomial approximation is applied.

### 2.3. Procedure for the Performance Investigation of Methods 

A general scheme of the test procedure for a dispersion curve reconstruction algorithm based on signals measured at two different distances is shown in Figure 1. Using the obtained signals, the segments of the Lamb wave phase velocity dispersion curves are reconstructed by both 2D-FFT method and the proposed algorithm.

Using the presented procedure, theoretical modelling is performed using the 2 mm thick aluminium plate. The suitability of the proposed method is evaluated in relation to the 2D-FFT method.

### 2.4. Numerical Investigation of the Methods

In order to show the proposed technique performance, it is applied for analysis of the simulated B-scan data. The numerical verification is performed using simulated signals of the Lamb waves propagating in the homogeneous aluminium plate of 2 mm thickness; the elastic parameters are: density *ρ* = 2780 kg/m^3^, Young modulus *E* = 71.78 GPa, and Poisson‘s ratio *ν* = 0.3435. The Lamb wave signals of the A_0_ and S_0_ modes are obtained according to [39]
(10)$$u_{x}\left(t\right)=\mathrm{IFT}[\mathrm{FT}[u_{0}\left(t\right)]\cdot H\left(jf,x\right)],$$
where $u_{x}\left(t\right)$ is the output signal, and $u_{0}\left(t\right)$ is the input signal. The IFT denotes the inverse Fourier transform, $H\left(jf,x\right)$; the complex transfer function of the object given by $H\left(jf,x\right)=e^{-\alpha \left(f\right)x}e^{-j\omega \frac{x}{c_{ph}\left(f\right)}}$; *x* the propagation distance; $\alpha \left(f\right)$ the frequency-dependent attenuation coefficient; $c_{ph}\left(f\right)$ the phase velocity dispersion curve corresponding to the particular guided wave mode; *ω* the angular frequency; *j* is the basic imaginary unit $j=\sqrt{-1}$. As the Lamb wave attenuation in the case of a metal plate is very low, this parameter is eliminated, thus signal waveforms are only affected by the dispersion effect.

The incident signal $u_{0}\left(t\right)$, having a frequency of 300 kHz and a 3-period harmonic burst with a Gaussian envelope, is used for investigation. The transmitted signal waveform is calculated along with the distance of 200 mm with d*x* step of 0.1 mm separately for both A_0_ and S_0_ modes. Likewise, two B-scans for both modes with 2001 simulated signals are obtained. The B-scans of the Lamb wave A_0_ and S_0_ modes are presented in Figure 2a,b, respectively.

Then, using the described peak detection technique of 2D-FFT spectrum magnitude, the phase velocity segments for both modes in numerical estimates are obtained. The 2D-FFT spectrum magnitude with the reconstructed A_0_ and S_0_ modes phase velocities, by calculating the peak values of spectrum magnitude at particular frequencies, is presented in Figure 3a,b.

The comparison of the obtained results demonstrated a good coincidence Figure 3a,b. Consequently, the method is chosen as a reference method for the algorithm of two measured signals as a suitability estimation.

Using the B-scan images described above (Figure 2), the A_0_ and S_0_ modes phase velocity segments at particular frequencies are reconstructed, applying the algorithm of two signals analysis. The basic condition that the time of flight should be measured between the same phase points of the signals is determined and presented in [21]. Studies have shown that, to avoid errors, the different distances between two signals should be determined in the different dispersion levels. The signal delay distance between *x*_1_ and *x*_2_ should be not greater than the signal period in the region of the significant dispersion [21]. Therefore, at first, the determination of the appropriate distances should be conducted.

Knowing the phase and group velocities dispersion curves, the distance between two points is calculated according to:(11)$$\Delta x\leq \frac{T}{2}\frac{\left(c_{ph\cdot }c_{gr}\right)}{\left|c_{gr}-c_{ph}\right|}$$
where *T* is the period of the signal, and $c_{gr}$ is the group velocity of the modes of Lamb waves in the frequency range during the analysis.

Parameters used for the phase velocity dispersion curve reconstruction: five filters with 100 kHz bandwidth in the frequency range of 200–400 kHz. The distances between two spatial positions for the A_0_ mode ∆*x* = 0.3 mm and S_0_ ∆*x* = 0.8 mm are selected. The segments of the A_0_ and S_0_ modes phase velocity dispersion curves reconstructed using the algorithm of two signals analysis and with the 2D-FFT method are presented in Figure 4.

In order to evaluate the presented method possibility, the mean relative error $\delta_{c_{ph}}$ is used for the comparison of the results obtained by both methods: (12)$$\delta_{c_{ph}}=100\%\cdot \frac{1}{L}\sum_{l=1}^{L}\frac{\left|c_{phl}-c_{ph}^{\mathrm{FFT}}\right|}{c_{ph}^{\mathrm{FFT}}},$$
where $c_{phl}$ is the phase velocity points in a segment of the experimentally reconstructed dispersion curve; L is the total number of points in a segment; and $c_{ph}^{\mathrm{FFT}}$ is the phase velocity values calculated using the 2D-FFT method at the same frequency.

According to Equation (12), the mean relative error for the A_0_ mode $\delta_{c_{ph}}=0.51\%$ and for the S_0_ mode $\delta_{c_{ph}}=0.07\%$ is estimated. The calculation of this parameter confirms the presented algorithm’s suitability to estimate the phase velocity dispersion in the simple case.

The possibilities to evaluate the phase velocity dispersion of Lamb waves propagating in different complexity objects using the proposed algorithm are presented in the next section.

## 3. Experimental Research

### 3.1. Example of the Polyvinyl Chloride Film

The PVC film, with a thickness *d* = 200 µm and lateral dimensions 210 × 297 mm^2^, is used for the experimental study. The manufacturer provides the elastic constants of the PVC film as: Young’s modulus *E* = 2.937 GPa, Poisson’s ratio *ν* = 0.42, density *ρ* = 1400 kg/m^3^ (Vintec^®^Clear PVC, Scranton, PA, USA) [40].

The ultrasonic measurement system ‘Ultralab’ developed at the Ultrasound Research Institute of Kaunas University of Technology is used for experimental research. The PVC film is fixed in the PVC film-mounting bracket. The positioning of the transducers is performed using the scanner ”Standa” 8MTF-75LS05 (Standa Ltd., Vilnius, Lithuania). The contact-point type transducers with 180 kHz resonant frequency are exploited (Figure 5). In order to receive the recorded signal at different spatial distances with a clearly expressed envelope maximum, the transmitter is excited by three periods of sinusoidal signal possessing Gaussian shape and an amplitude of 500 V.

In total, 300 signals of each A_0_ and S_0_ modes are experimentally collected. Amplitudes of the measured signals are normalised to the maximal value and are colour coded, as shown in the B-scan image Figure 6a. The receiver is moved in the range from *x*_1_ = 57 mm to *x*_2_ = 87 mm relative to the fixed transmitter with the step of d*x* = 0.1 mm. The recorded signals of both modes at distance *x*_s_ = 70 mm from the excitation point, and its frequency spectrum are presented in Figure 6b,c respectively.

Two different modes signals are clearly visible in the B-scan image Figure 6a. The filtration, using a moving time window, is used to separate these modes. The boundaries of these windows are shown in Figure 6a by dotted lines. The separated A_0_ and S_0_ modes are normalised according to the maximum amplitudes.

The SAFE method is used, and the theoretical phase velocity dispersion curves over a wide frequency range are calculated Figure 7a. To apply an algorithm of two signals, the required parameters must be determined. The frequency band of filters is determined according to the received signal at 6 dB level (Figure 6c). The frequency range 160–196 kHz and 4 filters with 25 kHz bandwidth are selected for the investigation. The distance between two points is calculated by applying Equation (11). The distances ∆*x*_A_ = 0.3 mm and ∆*x*_S_ = 3 mm for the A_0_ and S_0_ modes are chosen respectively. The reconstructed segments of the phase velocity dispersion curves of the A_0_ and S_0_ modes using the proposed method with the SAFE calculations are presented in Figure 7b. A very good coincidence of the A_0_ mode reconstructed dispersion curve segment is obtained, but in the case of the S_0_ mode, a large discrepancy is seen (Figure 7b). In order to find out why such discrepancy occurs*,* additional measurements need to be carried out.

The film thickness is measured using the micrometre (ATORN, Hommel Hercules, Viernheim, Germany) by the entire perimeter of the film during the under test with 20 mm steps. The measurement accuracy of the micrometre is ±5 µm. The set thickness is *d* = 180 ± 5 µm, and it differs from that provided by the manufacturer (*d* = 200 µm). The density of the film is obtained by weighing it with precision scales and using the results of its thickness measurements. The density is calculated at ρ = 1340 kg/m^3^. Then, Young’s module is calculated at *E* = 3.82 GPa, and it differs from that provided by the manufacturer (*E* = 2.937 GPa). The obtained results indicate that the specification provided by the manufacturer might be inaccurate.

Therefore, a verified 2D-FFT method is used to obtain the dispersion curves of the Lamb waves. The A_0_ and S_0_ modes in numerical values are obtained using the described peak detection technique of 2D-FFT spectrum magnitude. The obtained segments of the A_0_ and S_0_ modes using the 2D-FFT method are presented in Figure 8a, and the obtained numerical values and calculations by SAFE are presented in Figure 8b.

The obtained results using the 2D-FFT method confirm that the specification provided by the manufacturer is inaccurate (Figure 8b). The numerical results obtained by the 2D-FFT method spectrum magnitude are used for further investigation. In order to obtain the distance between two adjacent signals, a wavelength-depended parameter is used. In this case, new calculations of the middle frequency band and phase velocity value should be performed. Thus, using the determined frequency range 160–196 kHz (Figure 6c), the middle of the frequency band *f*_md_ = 178 kHz is determined. This parameter corresponds to 178 kHz in the frequency range of the calculated dispersion curves by the 2D-FFT method (Figure 8b). Then, the phase velocity values are determined: *c*_phA_ = 313 m/s and *c*_phS_ = 1857 m/s for the A_0_ mode S_0_ modes, respectively. Using determined phase velocity values in the particular frequency range, the wavelengths for different modes are calculated: *λ*_A_ = *c*_phA_/*f*_md_ = 1.8 mm and *λ*_S_ = *c*_phS_/*f*_md_ = 10.3 mm for the A_0_ and S_0_ modes, respectively. The wavelengths are used to determine the optimal distance range suitable to use for the two adjacent signals. By varying the spatial distance Δ*x* between the acquired and analyzed signals, the deviation of the obtained phase velocities from the values obtained by the 2D-FFT method (Equation (12)) at the corresponding frequencies is calculated. The deviations of the phase velocity values from the values obtained by the 2D-FFT method for the A_0_ and S_0_ modes are presented in Figure 9a,b respectively.

The *x*-axis shows the distance between the signals Δ*x* normalised to the corresponding wavelength *λ*, and the *y* axis indicates the relative error $\delta_{c_{ph}}$ (Figure 9a,b). Thus, based on the presented results, the optimal distance between two adjacent signals is determined. The range of 0.17–0.5 wavelength (the distance between spatial positions of signals registration Δ*x =* 0.3–0.9 mm), where $\delta_{c_{ph}}$ is not higher than 5%, is appropriate to use in the case of the A_0_ mode (Figure 9a). Meanwhile, the smallest errors of only 1.2%, in the distance greater than 1.5 wavelengths (the distance between spatial positions of signals registration is greater than Δ*x =* 15.5 mm), are obtained in the case of the S_0_ mode (Figure 9b). A higher relative error in the case of the A_0_ mode is obtained due to the presence of the significant dispersion within the frequency range under investigation. According to the presented calculations of the wavelength deviations, the two distances for the A_0_ mode *x*_1_ = 70 mm and *x*_2_ = 70.5 mm (Δ*x*/*λ* = 0.28) (Figure 9a), and the S_0_ mode *x*_1_ = 61 mm and *x*_2_ = 79.7 mm (Δ*x*/*λ* = 1.8) (Figure 9b) are chosen to reconstruct the dispersion curves. The reconstructed dispersion curves by the method of two signals analysis together with 2D-FFT method are presented in Figure 10.

Based on the obtained results (Figure 10), the mean relative error $\delta_{c_{ph}}$ is calculated according to Equation (12) for both modes. The mean relative error for the A_0_ mode (distances *x*_1_ = 70 mm and *x*_2_ = 70.5) $\delta_{c_{ph}}=2.8\%$, and for the S_0_ mode (distances *x*_1_ = 61 mm and *x*_2_ = 79.7 mm) $\delta_{c_{ph}}=0.99\%$ is determined.

### 3.2. Example of the Wind Turbine Blade

The second sample used in the study is the complex multilayer structure—the wind turbine blade (WTB). The whole WTB consists of several layers: surface layer (polymer paint, thickness 0.5 mm and GFRP skin, thickness 2 mm), GFRP main spar of 18.5 mm, and epoxy glue of 1 mm (Figure 11). The GFRP plies orientation in the main spar of WTB and skin is +45°/−45° and 0°/90°/+45°/−45°/0°, respectively. The general parameters of this multilayer structure that would describe the whole object are not specified. The analytical and/or semi-analytical methods could not be used, the 2D-FFT method is applied.

The experimental study is performed using the same wideband ultrasonic transducers with 180 kHz resonant frequency, developed at the Ultrasound Research Institute. The frequency bandwidth of the contact transducer is from 40 kHz up to 640 kHz (at −10 dB) [41]. The same ultrasonic measurement system ‘Ultralab’, as in the previous study, is used. The positioning and scanning of the transducers were performed using the linear mechanical scanner. The experimental set-up of the wind turbine blade testing is presented in Figure 11.

In order to excite the transmitter in low-frequency mode (from 16 kHz up to 90 kHz) and to achieve low reverberations in the time domain, the single pulse of 10 µs duration and amplitude of 100 V is used. The initial distance *x*_1_ between the transmitter and the receiver is 83 mm. The receiver is scanned at a distance from the transmitter along the region of the main spar of WTB. The final scanning distance is *x*_2_ = 533 mm, the entire scanning distance is 450 mm, and the scanning step is *dx* = 1 mm.

The B-scan image obtained during the experiment is shown in Figure 12a. The figure below clearly demonstrates the high-amplitude A_0_ mode signals, while the S_0_ mode signals are not clearly distinguishable. Therefore, filtration using a moving time window is used to separate the A_0_ mode. The boundaries of this window are displayed in Figure 12a by dotted lines. A signal at the distance *x*_s_ = 200 mm and its frequency spectrum are presented in Figure 12b,c, respectively.

Since only the A_0_ mode is obviously distinguished in the B-scan image (Figure 12a), the segments of the phase velocity dispersion curve are reconstructed only for the A_0_ mode. It can be seen from the A_0_ spectrum, as shown in Figure 12c, that the two frequency ranges are visible. The results obtained using the 2D-FFT method also clearly show two different frequency ranges (Figure 13a). The 2D-FFT spectra and the reconstructed phase velocities of the A_0_ mode calculating the peak values of spectrum magnitude from 2D-FFT are presented in Figure 13a,b, respectively.

Based on the obtained results by 2D-FFT method (Figure 13b), two frequency ranges are used for the phase velocity dispersion curve reconstruction by applying method of two signals analysis. The first frequency range is within 22–38 kHz; in this case, seven filters with 6 kHz bandwidth are selected for the investigation, and the middle value of the frequency band *f*_md_ = 30 kHz is determined. The second frequency range is within 57–64 kHz, four filters with the 6 kHz bandwidth are used, and the middle value of the frequency band *f*_md_ = 60 kHz is determined. The dispersion curves obtained by 2D-FFT method (Figure 13b) are used for the determination of the phase velocity values: the *c*_phA1_ = 1200 m/s in the 22–38 kHz frequency range and *c*_phA2_ = 1264 m/s in the 57–64 kHz frequency range are determined. Then, the wavelengths are calculated *λ*_A1_ = 40 mm and *λ*_A2_ = 21.1 mm for the corresponding frequency ranges. By varying the spatial distance between the acquired and analysed signals Δ*x*, the deviation of the obtained phase velocities from the values obtained by the 2D-FFT method (Equation (12)) at the corresponding frequencies are calculated. The obtained deviations of the phase velocities from the values obtained by the 2D-FFT method in different frequency ranges are presented in Figure 14a. The mean relative error $\delta_{c_{ph}}$ is calculated separately for both selected frequency ranges. The range of 0.18–0.38 wavelength (the distance between spatial positions of signals registration Δ*x =* 7–15 mm, curve No.1, Figure 14a) The range of 0.1–0.39 wavelength (the distance between spatial positions of signals registration Δ*x =* 2–8 mm, curve No.2, Figure 14a), where $\delta_{c_{ph}}$ is not higher than 3% (Figure 14a), is appropriate to be used as the optimal distance between two adjacent signals in both investigated frequency ranges.

According to the presented calculations of the wavelength deviations of the A_0_ mode, the set of a few distances was estimated. For the first case, the distances of *x*_1_ = 225 mm and *x*_2_ = 237 mm (Δ*x*/*λ* = 0.3) were estimated for the frequency range of 22–38 kHz (Figure 14a). For the second case, the distances of *x*_1_ = 225 mm and *x*_2_ = 231 mm (Δ*x*/*λ* = 0.3) were estimated for the frequency range of 57–64 kHz (Figure 14a).

The reconstructed dispersion curves of the phase velocities using the proposed algorithm (using estimated frequency ranges and distances of signal analysis) and by calculating peak values of spectrum magnitude from 2D-FFT are presented in Figure 14b.

Based on the obtained results Figure 14b the mean relative error $\delta_{c_{ph}}$ is calculated according to Equation (12) for both cases of frequency ranges of the A_0_ mode separately. The mean relative error in the range 22–38 kHz (distances *x*_1_ = 225 mm and *x*_2_ = 237 mm) is $\delta_{c_{ph}}=1.3\%$ and in the range 57–64 kHz (distances *x*_1_ = 225 mm and *x*_2_ = 231 mm) is $\delta_{c_{ph}}=0.3\%$. The total calculated mean relative error $\delta_{c_{ph}}=0.8\%$ is determined. Summarizing, it can be stated that the method of two adjacent signals analysis is suitable to be used for the evaluation of the Lamb wave phase velocity dispersion in objects with complex construction.

## 4. Conclusions

The article analyses the possibilities of the method of two adjacent signals to reconstruct the phase velocity dispersion curves of Lamb waves propagating in two different types of objects. The non-homogeneous polyvinyl chloride (PVC) film and complex object of the wind turbine blade (WTB) were selected for the research. In order to obtain the numerical estimates, a comparison with a new algorithm based on peaks of spectrum magnitude was presented and analyzed. The wavelength-dependent parameter was proposed to be used for determining the optimal distance range, which is necessary to be used for the method of two adjacent signals application. It was determined that the range of 0.17 ÷ 0.5 wavelength, where $\delta_{c_{ph}}$ is not higher than 5%, is appropriate to be used in the case of the A_0_ mode in PVC film sample. The smallest error of 1.2%, at a distance greater than 1.5 wavelengths, were determined in the case of the S_0_ mode. By using distances *x*_1_ = 70 mm and *x*_2_ = 70.5 mm for the A_0_ mode (Δ*x*/*λ* = 0.28) and *x*_1_ = 61 mm and *x*_2_ = 79.7 mm for the S_0_ mode (Δ*x*/*λ* = 1.8), the dispersion curves are reconstructed in the sample of polyvinyl chloride (PVC) film.

A comparison of the results with the 2D-FFT algorithm based on peaks of spectrum magnitude was performed, and the mean relative error $\delta_{c_{ph}}=2.8\%$ for the A_0_ mode and for the S_0_ mode $\delta_{c_{ph}}=0.99\%$ was determined. In the case of WTB sample, the range of 0.18 ÷ 0.39 wavelength, where $\delta_{c_{ph}}$ is not higher than 3%, was used as the optimal distance range between two adjacent signals. Using distances *x*_1_ = 225 mm, *x*_2_ = 237 mm (frequency range of 22–38 kHz, Δ*x*/*λ* = 0.3) and *x*_1_ = 225 mm and *x*_2_ = 231 mm (frequency range of 57–64 kHz, Δ*x*/*λ* = 0.3) of the A_0_ mode, the two segments of the dispersion curves were reconstructed. The comparison of the results with 2D-FFT algorithm based on peaks of spectrum magnitude was performed, the mean relative error $\delta_{c_{ph}}=1.3\%$ for first reconstructed range (22–38 kHz) and $\delta_{c_{ph}}=0.3\%$ for second range (57–64 kHz) was determined. The total mean relative error $\delta_{c_{ph}}=0.8\%$ was calculated. The obtained results indicate that the method of two adjacent signals analysis is appropriate to be used for the evaluation of the Lamb wave phase velocity dispersion in non-homogeneous objects and objects with complex construction.

## Figures and Tables

**Figure 1 materials-14-06990-f001:**
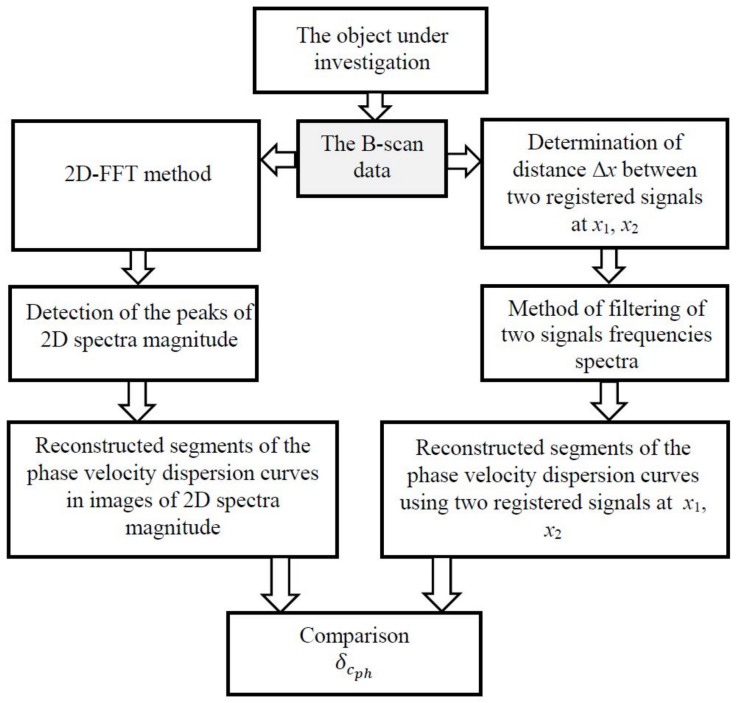
A general scheme of the test procedure for a dispersion curve reconstruction algorithm based on signals measured at two different distances.

**Figure 2 materials-14-06990-f002:**
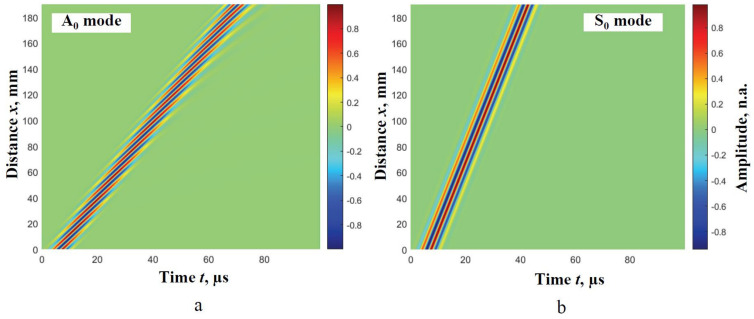
B-scan images of the simulated wave A_0_ (**a**) and S_0_ (**b**) modes of Lamb waves at 300 kHz frequency.

**Figure 3 materials-14-06990-f003:**
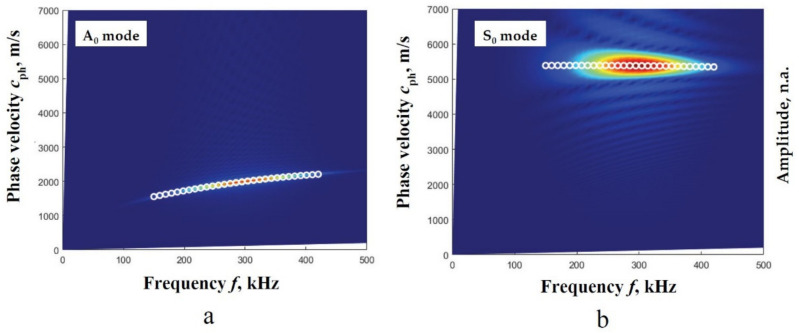
The 2D-FFT spectrum magnitude (presented by the color picture) with the reconstructed phase velocities (presented by white circles) of the A_0_ (**a**) and S_0_ (**b**) modes by calculating the peak values of spectrum magnitude at particular frequencies.

**Figure 4 materials-14-06990-f004:**
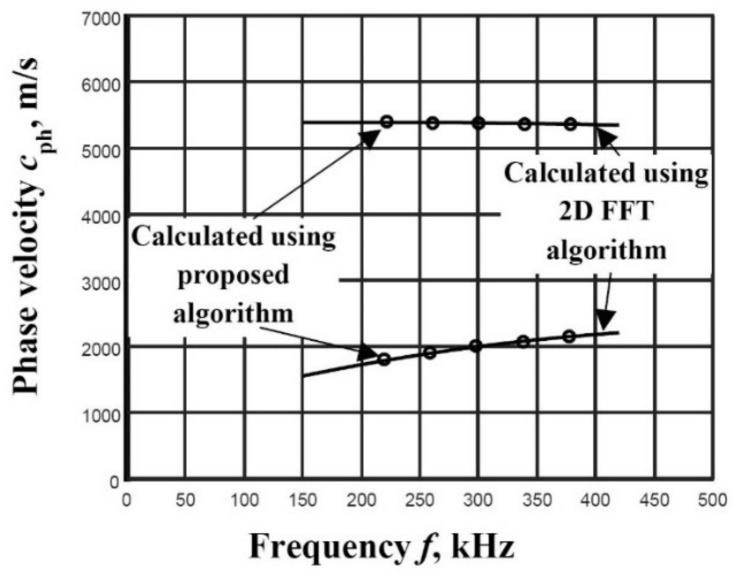
The reconstructed phase velocities of the A_0_ and S_0_ modes using the proposed algorithm (presented by circles) and by calculating peak values of spectrum magnitude from 2D-FFT (presented by solid line).

**Figure 5 materials-14-06990-f005:**
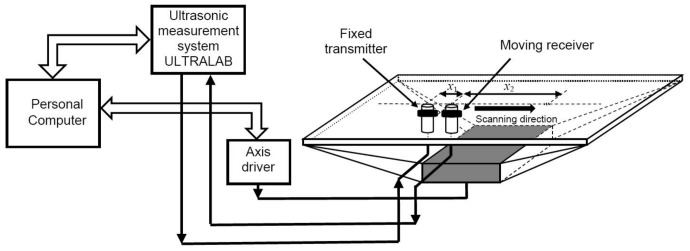
Experimental setup of the generation and recording of the Lamb waves signals propagating in PVC film.

**Figure 6 materials-14-06990-f006:**
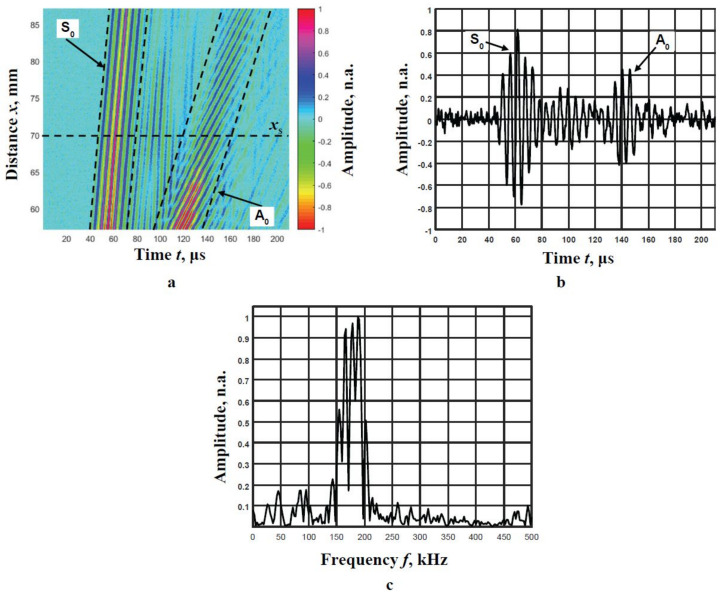
The B-scan image with the collected A_0_ and S_0_ modes signals (color coded) (**a**), the waveform of the signal at the distance of *x*_s_ = 70 mm from the excitation point (**b**) and frequency response of this signal (**c**).

**Figure 7 materials-14-06990-f007:**
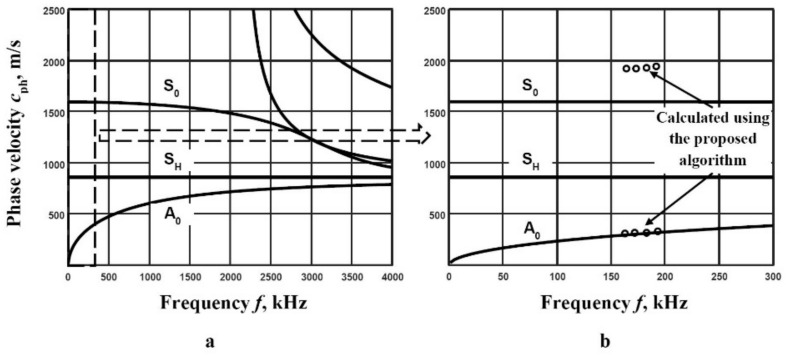
Phase velocity dispersion curves of Lamb wave modes calculated by the SAFE method in *d* = 200 µm thickness PVC film (**a**) and the reconstructed segments of the phase velocity dispersion curves of the A_0_ and S_0_ modes, using the proposed method (dots) and the SAFE calculations (**b**).

**Figure 8 materials-14-06990-f008:**
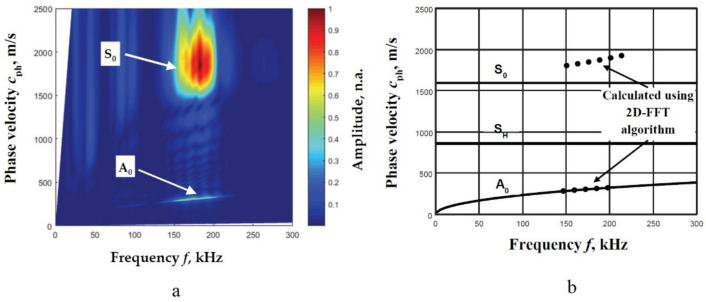
The 2D-FFT spectrum magnitude (colour) (**a**) and the reconstructed phase velocities of the A_0_ and S_0_ modes by estimation of the peak values of spectrum magnitude from 2D-FFT and SAFE calculations (**b**).

**Figure 9 materials-14-06990-f009:**
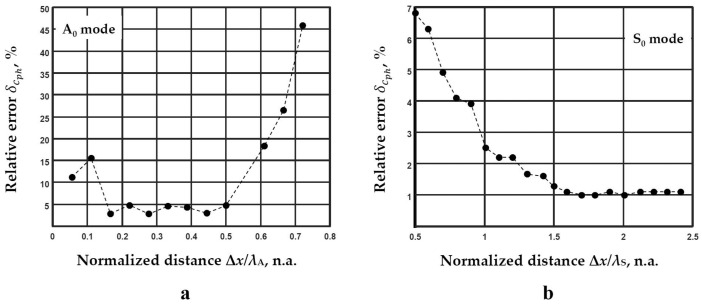
The deviation of the obtained phase velocities from the values calculated by the 2D-FFT method for the A_0_ mode (**a**) and S_0_ modes (**b**).

**Figure 10 materials-14-06990-f010:**
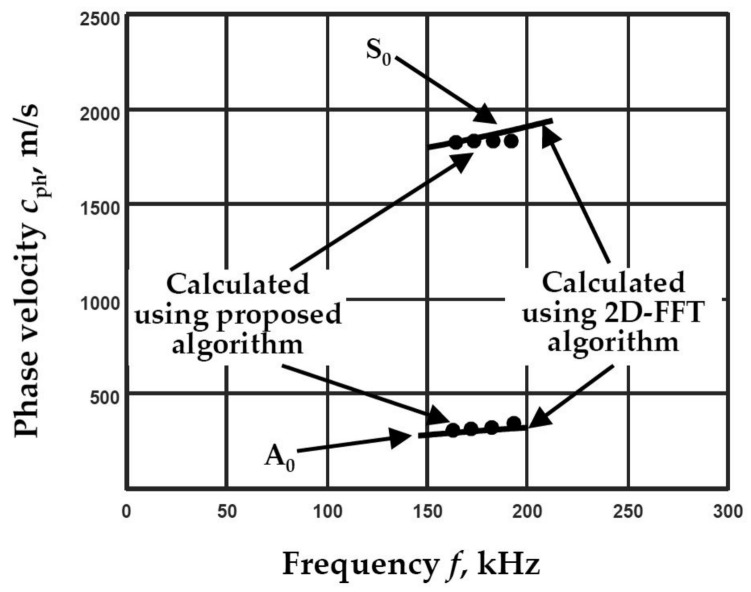
The.reconstructed phase velocities of the A_0_ and S_0_ modes using the proposed algorithm (dots) and by calculating peak values of spectrum magnitude from 2D-FFT (line).

**Figure 11 materials-14-06990-f011:**
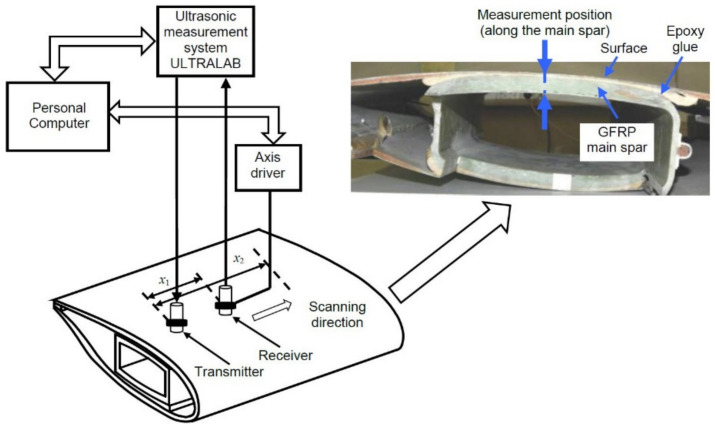
The experimental setup of the generation and registration of the Lamb waves signals propagating in a wind turbine blade.

**Figure 12 materials-14-06990-f012:**
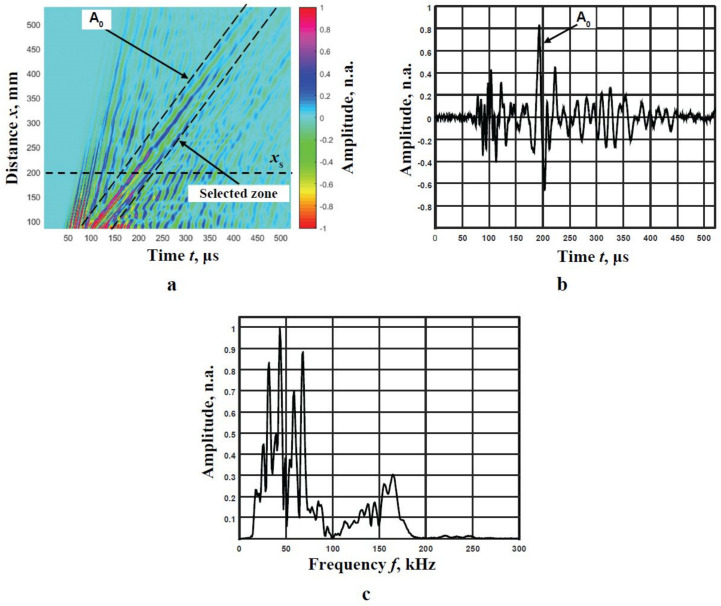
The B-scan with the A_0_ and S_0_ modes (**a**), the waveform of the signal at the 200 mm distance from the excitation point (**b**), the frequency response of this signal (**c**).

**Figure 13 materials-14-06990-f013:**
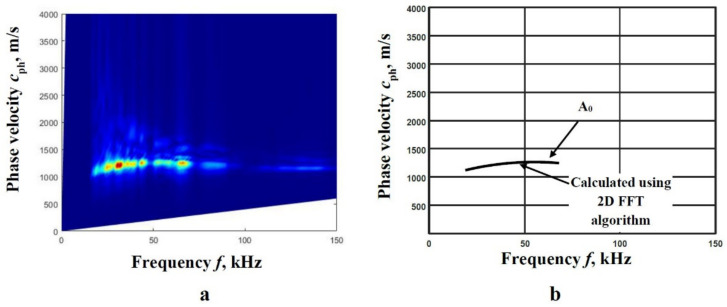
The 2D-FFT spectra (colour) (**a**), the reconstructed phase velocities of the A_0_ mode calculating the peak values of spectrum magnitude from 2D-FFT (**b**).

**Figure 14 materials-14-06990-f014:**
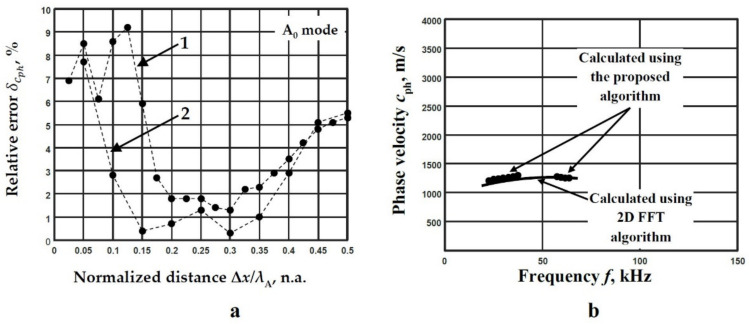
The deviation of the obtained phase velocities from the values obtained by the 2D-FFT method in different frequency ranges: 1—22–38 kHz, 2—57–64 kHz (**a**), the reconstructed dispersion curves of the phase velocities using the proposed algorithm (dots) and by calculating peak values of spectrum magnitude from 2D-FFT (line) (**b**).

## Data Availability

The data presented in this study are available on request from the corresponding author.

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
