# Peer review of "Reconstruction of Lamb Wave Dispersion Curves in Different Objects Using Signals Measured at Two Different Distances"

_materials, 2021, doi:10.3390/ma14226990_

Round 1

Reviewer 1 Report

The paper presents a method for estimating dispersion curves based on measurements at two locations. While the application is of interest for NDE and material characterization purposes, it is necessary to address the following issues before considering the paper for publication:

1- It is not clear what advantage the proposed technique has over the traditional frequency domain analysis where two signals measured along the direction of wave propagation are related, in the frequency domain, as follows:

U(f)_2=U(f)_1*exp(-ik(x2-x1)), where k is the wavenumber

This technique also estimates dispersion curves based on two measurements only and has the advantage of not being dependent on filtering, envelop construction, and other approximations introduced by the authors of this manuscript.

Ref.  J.F. Doyle, Wave Propagation in Structures: Spectral Analysis Using Fast Discrete Fourier Transforms, Springer, 1997.

2- When an envelope is used to describe a signal, it is the group velocity that naturally comes to mind. The authors are using envelops for phase velocity estimation, this needs to be further explained.

3- It is not clear how this method handles reflections and the presence of different wave modes (like overlapping A0 and S0 wave forms).

4- There are several parameters and empirical relations introduced by the authors for this technique to work (such as the filter being used, the required distance between measurement locations, the way frequency is calculated, …). Such parameters appear to be case-dependent; thus the technique will require some fine-tuning every time a new system is inspected. This fine-tuning depends on knowing the answer beforehand, which defeats the purpose of this work. A thorough sensitivity analysis is needed to better explain the dependence on these parameters if this work is to be generalized.

5- Measurement locations used in this analysis are very closely spaced (0.5 mm). How does uncertainty in this distance affect the results? Is a micrometer level of accuracy is required?

6- On a less-technical note:

a- The introduction over discusses the materials being investigated in the study (like PVC). Such a discussion distracts from the main point the paper is trying to make.

b- At several places, unclear, incoherent, and broken sentences are found. The paper needs to be revised for clarity and grammatical correctness.

Reviewer 2 Report

The paper is an improved version of a previous submission. In the previous review, the reviewer had pointed out that the lack of novelty is the major argument against the publication of the paper.

The work is merely a use case of an algorithm developed before. It does not add any significant insight or valuable knowledge in the eyes of the reviewer. As a result the reviewer will recommend rejection.

The method should be extended to include uncertainty studies or some other novelty such as determination of the optimal distance between the scanned points. 

Reviewer 3 Report

Dispersion curves extraction is one of the fundamental and classical topics in the field of ultrasonic guided waves based NDT and NDE. Many representative methods have been reported. The present work aims to extract the dispersion curves using two signals measured at different distances, which can be a feasible method for low-cost and efficient processing in some NDT application. The major weakness of the manuscript is the writing, (1) authors mixed the results, experimental setup, method together, which bring challenges for readership, (2) several advanced signal processing methods for dispersion curves extraction should be added. The authors may want to consider the following comments.

[1] It should be pointed out that the phase velocity dispersion curves can certainly be well extracted within the low-attenuation shell-like structure. Several literatures should be properly cited. For instance, recent progresses using the phase array have allowed the dispersion curves extraction and parameter inversion for the high attenuation waveguide, including composite material(ULTRASONICS, 2021, 115, pp. 106427)and biological tissue, such as the long cortical bone using the sparse-SVD method. Some advanced signal processing for the dispersion curves extraction and mode extraction of single channel data could also be considered, such as the dispersive Radon transform(J ACOUST SOC AM, 2018, 143, (5), pp. 2729-2743)

[2] Please separate the results and method section. For instance, Fig. 1 and Fig. 2 are results, which should not appear in the method section. A possible solution can be to combine Fig. 1 and 2 with Fig, 4. In addition, Fig. 4 should belong to results. It’s improper to mix it with the method section.

[3] The experimental setup can be an individual section. Fig. 10 should be in the experimental section also rather than the results part.

[4] It should be pointed out that due to the finite aperture limit, SNR and attenuation, the classical 2D-FT method cannot provide very accurate estimates of the dispersion curves. Some advanced signal processing such as the Radon transform and Synchrosqueezed wavelet transform can be helpful for high resolution extraction of the dispersion curves.

[5] Eq. (3), please provide full word for IFT

[4] Eq. (5), ??1,?? and ??2,?? are the zero-crossing time instants, please clarify how to determine the zero-crossing time? Any influence on the final results, when using unproper thresholding?

[6] Eq. (6-7), out of the page, please revise

[7] A diagram of the algorithm can be helpful, if possible. The number before each paragraph from line 114 to 141 may not be so necessary.

[8] Eq. (2) Bk = exp(4ln(0.5)…), please explain why there is a coefficient of 4ln(0.5)?

Reviewer 4 Report

The authors, in their previous work, proposed an algorithm for Lamb wave phase velocity dispersion curves estimation using signals measured at two adjacent locations. In this study however, authors tend to study reliability and susceptibility of the previously proposed technique based on two test cases. From that, the experiments and results are very limited. In my opinion, in order to properly test the reliability and susceptibility of the new method more test cases are required. For each specimen a number of acquisitions for different positions could be analyzed in order to properly investigate robustness of the method.

The paper is well written and should be of interest to the readership of Materials, but it should be greatly improved before publishing.

Other comments:

Statistical analyzes should be performed.

Drawbacks and limitations of the method should be discussed.

How was the theoretical phase velocity, A0 in Eq. (12) and S0 in Eq. (13), estimated for the PVS and WTB? What if these values are not known a priori as is the case with most applications? Can you still use the proposed method reliably? This should be discussed in the text. 

Can this method be applied to higher order modes?

It seems that the proposed algorithm works for nondispersive (or with very limited dispersion) frequency ranges. Does it also work for frequencies where Lamb wave modes are more dispersive?

Line 374: How did you calculate the density?

Round 2

Reviewer 2 Report

The paper is an improved version of previous submission.

At the previous round, the paper was recommended for rejection due to the lack of novelty and just an application of existing technique for new samples. As a solution the authors have put in statistical metrics for the comparison of the performance.

The authors indeed have increased the impact of the paper, but the scientific rigour of the paper is somewhat limited in the current state.

For instance the error obtained for the 2 samples is significantly different, also, the shape of the error with ratio is slightly different for the two samples. So an investigation into the reason for this difference needs to be presented.

The error may be due to systematic errors or measurement errors so they need to be identified and a clear discussion included. 

The higher error in S0 mode, the authors claim additional studies are necessary, but it is important to identify the reason for the additional error. In general S0 wave being less dispersive the error is expected to be lower than the A0 wave. so some additional discussion is needed.

In general the paper flow is still not changed from the previous version. The line 100-106 which identify the main contribution of the paper need to be edited to be in line with the conclusions section.

Reviewer 3 Report

The reviewer appreciates authors' efforts during the revision. The manusript has been improved. However, I still don't think the structure of the manusript is clear enough for the readership. Furthermore, the introduction is not well written and the authors fail to clarify the major original contribution of the present work.  

Round 3

Reviewer 2 Report

The authors have addressed all of my concerns

The paper may be accepted in current form

Reviewer 3 Report

The manuscript has been improved, and I think it can be accepted in present form. 

This manuscript is a resubmission of an earlier submission. The following is a list of the peer review reports and author responses from that submission.

Round 1

Reviewer 1 Report

This manuscript presents two examples where a previously developed algorithm, ref. 11 in the current manuscript, is used to estimate dispersion curves. Given that the algorithm used in this study has been presented and validated with “nonhomogeneous objects”, as stated by the authors (page 3, line 125), the contribution of the current manuscript is very marginal. It simply presents more of the same analysis as in the authors’ previous work (ref. 11). Based on this, the reviewer does not recommend this manuscript for publication at Materials.

Additional comments are listed below:

  1. The introduction section does not adequately cover the state of the art. Several methods for estimating dispersion curves are not discussed, (some of which do this with 2 points only, which makes them very relevant to this study). A more comprehensive literature review is required.
  2. Page 2, line 91: measurement locations are presented as a function of time, however, later in the manuscript waveforms are measured at fixed locations. This needs to be corrected.
  3. The description of the algorithm is not clear. This leaves the reader with several questions that might be answered in Ref. 11.
  4. The accuracy of the results seems to be strongly dependent on the distance between the two locations, as stated by the authors. While the authors give some guidelines for selecting this distance, it seems that some trial-and-error is involved in the process. A more thorough discussion on this point is needed where the sensitivity of this method to this parameter is evaluated.
  5. How would the algorithm work in the following scenarios:
    1. If the different wave modes cannot be separated using a moving time window.
    2. If the response measured at a given location has reflected waveforms overlapping with incident waves.  
  6. The discussion regarding updating PVC material properties is not necessary (common knowledge in a way).
  7. For the WTB, more than two wave modes exist at the frequency range selected in the study. It is not clear who the authors managed to only excite A0 and S0 modes.
  8. There are several typos, incoherent and broken sentences throughout the manuscript.

Best wishes,

Reviewer 2 Report

The paper deals with the evaluation of a method for reconstructing the dispersion curve based on two measurements. 

The authors have already published the findings of the measurements in [11] where the isotropic and anisotropic materials were presented. The current study shows the validity for PVC and wind turbine blade. 

The paper is fairly well written and the results are good. But the major problem is the novelty. The paper is more of an application of existing technique, so does not really add any new knowledge. So it is more fitting for a conference, than a journal.

The authors should somehow expand the study to incorporate more uncertainties, and futher apply the technique for real applications

Reviewer 3 Report

  1. As the authors stated, the algorithm of the proposed signal processing method was presented by Draudviliene et al. in the previous paper. This paper is to apply the proposed method to complex objects, which is "Incremental innovation". I suggest that the characteristics of non-homogeneous objects should be considered to improve the algorithm.
  2. As the authors declared, "research must be made to determine whether suitable signal processing methods are able to evaluate these named characteristics of the Lamb waves which would be simple and easy to use, but at the same time quickly provide the necessary information." The computation effort of the method should be provided and discussed.
  3. Some related work on the topic of dispersion curve extraction  can be cited in the Introduction part. Such as 
    1) Sparse SVD method for high-resolution extraction of the dispersion curves of ultrasonic guided waves, IEEE UFFC 2016
    2) High-resolution Lamb waves dispersion curves estimation and elastic property inversion, Ultrasonics 115, 106427
  4. English should be revised to facilitate comprehension; just to mention a few examples, lines 17-19, lines 85-86, lines 163-165, and so on. Please check the punctuation. For example, lines 159-160.
  5. Technological research requires accuracy and clarity, "necessary information" in line 21 should be specific.